# Spatial Heterogeneity and Temporal Trends in Malaria on the Thai–Myanmar Border (2012–2017): A Retrospective Observational Study

**DOI:** 10.3390/tropicalmed4020062

**Published:** 2019-04-12

**Authors:** Sayambhu Saita, Tassanee Silawan, Daniel M. Parker, Patchara Sriwichai, Suparat Phuanukoonnon, Prayuth Sudathip, Richard J. Maude, Lisa J. White, Wirichada Pan-ngum

**Affiliations:** 1Department of Tropical Hygiene, Faculty of Tropical Medicine, Mahidol University, Bangkok 10400, Thailand; phoori.ph@gmail.com; 2Department of Community Health, Faculty of Public Health, Mahidol University, Bangkok 10400, Thailand; tsilawan@gmail.com; 3Department of Population Health and Disease Prevention, University of California Irvine, Irvine, CA 92697, USA; dparker1@uci.edu; 4Department of Medical Entomology, Faculty of Tropical Medicine, Mahidol University, Bangkok 10400, Thailand; patchara.sri@mahidol.ac.th; 5Department of Social and Environmental Medicine, Faculty of Tropical Medicine, Mahidol University, Bangkok 10400, Thailand; suparatp@hotmail.com; 6Bureau of Vector-borne Diseases, Department of Disease Control, Ministry of Public Health, Nonthaburi 11000, Thailand; psudathip@gmail.com; 7Centre for Tropical Medicine, Nuffield Department of Medicine, University of Oxford, Oxford OX3 7BN, UK; Richard@tropmedres.ac (R.J.M.); Lisa@tropmedres.ac (L.J.W.); 8Mahidol-Oxford Tropical Medicine Research Unit, Faculty of Tropical Medicine, Mahidol University, Bangkok 10400, Thailand; 9Harvard T.H. Chan School of Public Health, Harvard University, Boston, MA 02115, USA

**Keywords:** spatial patterns, temporal trends, *Plasmodium falciparum*, *Plasmodium vivax*, Thai–Myanmar border

## Abstract

Malaria infections remain an important public health problem for the Thai–Myanmar border population, despite a plan for the elimination by the end of 2026 (Thailand) and 2030 (Myanmar). This study aimed to explore spatiotemporal patterns in *Plasmodium falciparum* and *Plasmodium vivax* incidence along the Thai–Myanmar border. Malaria cases among Thai citizens in 161 sub-districts in Thailand’s Kanchanaburi and Tak Provinces (2012–2017) were analyzed to assess the cluster areas and temporal trends. Based on reported incidence, 65.22% and 40.99% of the areas studied were seen to be at elimination levels for *P. falciparum* and *P. vivax* already, respectively. There were two clear clusters of malaria in the region: One in the northern part (Cluster I), and the other in the central part (Cluster II). In Cluster I, the malaria season exhibited two peaks, while there was only one peak seen for Cluster II. Malaria incidence decreased at a faster rate in Cluster I, with 5% and 4% reductions compared with 4% and 3% reductions in *P. falciparum* and *P. vivax* incidence per month, respectively, in Cluster II. The decreasing trends reflect the achievements of malaria control efforts on both sides of the Thai–Myanmar border. However, these clusters could act as reservoirs. Perhaps one of the main challenges facing elimination programs in this low transmission setting is maintaining a strong system for early diagnosis and treatment, even when malaria cases are very close to zero, whilst preventing re-importation of cases.

## 1. Introduction

Malaria infections threaten more than 3.2 billion of the world’s population, with approximately 216 million cases globally each year, of which 6.76% occur in Southeast Asia [1]. In the Greater Mekong Sub-region (GMS), which includes Cambodia, China (Yunnan Province), Lao PDR, Myanmar, Thailand, and Vietnam, malaria has been in general decline over the last several decades. All nations of the GMS have committed to eliminating malaria by the year 2030 [2,3]. Thailand has already made huge strides in reducing its malaria burden, with the disease persisting only along its international borders with Myanmar, Laos, and Cambodia. Historically, some of heaviest burdens of malarial disease have been in provinces along the Thai–Myanmar border [4,5].

There are several potential barriers to the elimination of malaria in the GMS. Malaria dynamics at border areas are driven by complex interactions between pathogen, vector, and human populations, which often link areas with different degrees of malaria transmission capacity, leading to continuous malaria transmission within endemic areas [6,7]. More recently, resistance to artemisinin combination therapies (ACT) has spread across the GMS [4,8], presenting a widespread threat of untreatable malaria. In the past, drug-resistant malaria has originated in the GMS and subsequently spread to Africa, resulting in humanitarian disasters [9,10]. The current threat has put pressure on nations within the GMS to rapidly eliminate malaria before it becomes completely untreatable, and before these strains spread to other parts of the world.

A strategy for malaria elimination in the GMS was developed by the World Health Organization (WHO), with the ultimate goal of eliminating all malaria in all GMS countries by 2030 and, considering the urgent action required to combat multidrug resistance in the region, to eliminate *Plasmodium falciparum* by 2025. WHO recommends that nations with reduced malaria burdens shift from pre-elimination to elimination strategies once they have achieved an incidence of less than 1 case per 1000 people per year [3]. Once a nation has achieved zero locally-acquired infections for a period of three years, it can be certified as malaria free (as has recently been the case in Sri Lanka, Uruguay, and other countries). The National Malaria Elimination Strategy of Thailand (2017–2026) has also set its own malaria elimination goals, including an increase of malaria free areas to more than 75% by 2017, 90% by 2020, and 100% by 2023, with morbidity and mortality to not exceed 0.20 per 1000 population and 0.01 per 100,000 population, respectively, by the year 2021 [11]. In order to achieve the goal, several malaria control and elimination strategies have been implemented in endemic areas along the Thai–Myanmar border i.e., insecticide-treated nets (ITNs), long-lasting insecticide-treated bed nets (LLINs) (adopted since 2008 [12]), indoor residual insecticide spraying (IRS) (adopted since 1953 [12]), rapid diagnostic tests (RDTs) (adopted since 2001–2002 [13] and at the same time as malaria posts in 2014 [11]), and ACTs (adopted since 1995 [12] and intensively implemented in both Thailand and Myanmar borders from 2014 [11,13,14,15]).

In most previous studies, malaria elimination projects were heavily focused on *P. falciparum* [16,17,18,19]. In order to truly eliminate malaria, it will be necessary to also focus on the other malaria species, of which *Plasmodium vivax* is the major contributor in the GMS. Thus, the present study aimed to explore the spatial distribution of both *P. falciparum* and *P. vivax* infections, detect clusters of infections along the Thai–Myanmar border, where the malaria burden has historically been concentrated, and to assess disease trends among Thai citizens in the individual sub-districts between 2012 and 2017.

## 2. Materials and Methods

### 2.1. Study Design and Study Areas

This study was a retrospective observational study using existing data collected at the sub-district (Tambon) level. The study area was located along the Thai–Myanmar border in Kanchanaburi and Tak Provinces (Figure 1), encompassing 22 districts and 161 sub-districts. The total combined population of both provinces was 1,532,250, with a district level population density ranging from 8.06 to 399.42 person/km^2^. The western edge of Kanchanaburi and Tak Provinces has a long boundary with Kayin and Mon States, and the Tanintharyi Region of Myanmar. Generally, the geographical characteristics of the Thai–Myanmar border are forest, mountain, and plantations, with little urbanization. The physical geography of this area includes watersheds, river basins, valleys, tertiary forests, and, occasionally, dense pockets of human settlements, ranging from refugee camps and agricultural villages, to river and border-trading towns [14,20]. There are diverse human and environmental factors in this area that are relevant to malaria ecology. In particular, the area is situated in a rain shadow zone, leading to agricultural and forest characteristics that might affect the conditions and suitability of vector breeding sites in these areas.

### 2.2. Malaria Data

The Thai Ministry of Public Health (MOPH) established an official malaria database in 2012. An algorithm was developed to effectively detect and delete data redundancy between two reporting systems operated by the Bureau of Epidemiology (BOE) and the Bureau of Vector Borne Diseases (BVBD). The BOE reporting system was hospital-based while the BVBD reporting system obtained weekly data from district vector-borne disease control units, malaria posts, and malaria clinics [21,22]. Both the BOE and BVBD agreed to combine data from these two different health information systems into a single national database [23]. Both the data consolidation algorithm and the process were approved by the MOPH and the information is now officially published through their website [24]. To clearly explain about data sources, malaria post is a community-directed health service unit that provides community outreach for malaria blood examination using RDT and prompt treatment using a recommended ACT for an individual infected in the pocket of endemic villages remotely located in a transmission control area, staffed by a community member so-called malaria post worker. Recently, RDT used in Thai–Myanmar border is CareStart™ Malaria HRP2/pLDH (Pf/pan) Combo which is based on detection of malaria antigens such as histidine-rich protein 2 (HRP2) for *P. falciparum* and pan malaria-specific antigen (pLDH) for all other malarial species. The sensitivity and specificity of the test for *P. falciparum* reported were 97.3% and 94.1%, respectively [25]. The limit of detection of RDTs was estimated to be around 100 parasites/μL, the densities at which, in low transmission settings, malaria infections are commonly presented with the signs and symptoms. Malaria clinics are staffed by a trained microscopist, with malaria diagnosis being confirmed using blood slides, followed by treatment for confirmed infections. Hospitals also use microscopy for diagnosis.

The present study focused on both *P. falciparum* and *P. vivax* malaria infections, which were diagnosed either by microscopy in hospitals and malaria clinics or RDT in malaria posts. The data were downloaded from the website in February 2018. Cases were reported by three different classifications of patient:Thai cases, which referred to cases among Thai citizens.M1 cases, which referred to cases reported in migrants who had been in Thailand for >six months.M2 cases, which referred to cases reported in migrants who had been in Thailand for <six months.

Our analysis predominantly focused on Thai citizens (for whom there exists a denominator obtained from annual government estimates), with an assumption that the malaria situation among Thai citizens was likely to be indicative of indigenous malaria (few Thais spend time in Myanmar). *P. falciparum* and *P. vivax* incidence rates were calculated (as number of cases per 1000 people per year) for Thai citizens and for all sub-districts. The annual estimates of Thai resident population by sub-district were obtained from the Department of Provincial Administration, Ministry of Interior, Thailand. Summary statistics and supplementary analyses include migrant malaria cases.

### 2.3. Spatial Distribution

Smoothed incidence rates for both *P. falciparum* and *P. vivax* were calculated using empirical Bayesian smoothing. The spatial empirical Bayesian (SEB) smoothed rate was used to map spatial distribution of annual *P. falciparum* and *P. vivax* incidence in of 161 sub-districts in Kanchanaburi and Tak Province from 2012 to 2017 for exploratory visualization. The smoothing method can be examined for solving the modifiable areal unit problems [26,27]. SEB smoothing technique uses empirical Bayes methods to borrow data on events from neighboring observations to minimize problems associated with small populations at risk and different population sizes [27]. Spatial weight matrices in this study were based on a queen contiguity matrix, which creates links between all neighbors sharing a common point or length on their boundaries [28]. SEB smoothed rate increases the ability to discern systematic patterns in the spatial variation of the outcome under study by reducing noise and making trends and patterns more obvious. This method has been widely used in the field of public health and spatial epidemiological study [29,30]. The SEB smoothed rates of Thai cases over mid-year population were calculated using GeoDaTM 0.9.5-I software. Then, those SEB smoothed rates were overlaid to the map using Quantum GIS version 2.14.20 software.

### 2.4. Spatial Clustering

Spatial autocorrelation among Thai citizens was investigated and quantified using the SEB smoothed rates and the Moran’s *I* and local indicators of spatial associations (LISA) statistics. The Moran’s *I* statistic is a measure of global spatial autocorrelation whereas the LISA statistic is a measure of local autocorrelation. Both global and local measures were used for each species (*P. falciparum* and *P. vivax*) and for each sub-district, for each year of the study period. Values of Moran’s *I* range from 1 to −1 and a large positive or negative Moran’s *I* indicates nearby areas are highly similar (or dissimilar), whereas Moran’s *I* near 0 indicates no spatial association in the data. The LISA statistics were investigated and mapped to identify local clusters (high–high or low–low) or local spatial outliers (high–low or low–high) of smoothed *P. falciparum* and *P. vivax* incidence at the sub-district level. Spatial weight matrices for quantifying spatial autocorrelation were based on a queen contiguity matrix [28]. The statistical significance was tested using 999 Monte Carlo permutations, and the Bonferroni method was used to account for multiple testing [31,32,33,34]. GeoDaTM version 0.9.5-I was used to calculate SEB smoothed rates, Moran’s *I,* and LISA statistics. All maps were made using Quantum GIS version 2.14.20 software.

Sub-districts that were identified as local clusters for three or more years during the study period study were further analyzed for temporal trends.

### 2.5. Seasonality and Trends

Trends and seasonality of monthly *P. falciparum* and *P. vivax* incidence among Thai citizens was assessed for each sub-district. A seasonal trend decomposition procedure based on Loess [35] (hereafter referred to as STL) was used to detect trends and seasonal variations in *P. falciparum* and *P. vivax* incidence. The general trends in incidence were then further assessed using Poisson regression [36], where the distributions of *P. falciparum* and *P. vivax* incidence for each cluster and individual sub-district were assumed to follow the Poisson distribution. Incidence rate ratios (IRR) and 95% confidence intervals (CI) were calculated to assess changes in *P. falciparum* and *P. vivax* incidence over the study period using a Poisson regression model. R statistical software version 3.4.3 was used for the seasonal and trend analyses.

## 3. Results

### 3.1. Summary Statistics

In Tak Province (Figure 1), total *P. falciparum* and *P. vivax* cases decreased by 98.10% and 86.41%, respectively, from 2012 to 2017. For both *P. falciparum* and *P. vivax*, the majority of cases were in individuals aged 5–14 years old; students (2013–2015); laborers (2016–2017); and Myanmar citizens. *P. falciparum* cases were most common among Thais and migrants while *P. vivax* cases were most common among Thais.

In Kanchanaburi Province (Figure 1), total *P. falciparum* cases decreased by 99.07% between 2012 and 2017. Most *P. falciparum* cases were seen in individuals aged 25–44 years old, agricultural workers, and M1 (long-term) migrants. *P. vivax* cases in Kanchanaburi Province decreased by 88.52% over the same time period. The majority of *P. vivax* cases were in individuals aged 5–14 years old (2013, 2014, and 2016) or 25–44 years old (2012, 2015, 2017); agricultural workers; Thai citizens (2012, 2015, 2016); and Myanmar citizens (2013, 2014, and 2017), especially M1 migrants. 

The health facilities in the study area that provided diagnoses (either by microscopy or RDT) and treatment included public hospitals, malaria clinics, and malaria posts [21] for *P. falciparum* and *P. vivax* cases of all classifications (i.e., Thai, M1, and M2) (Table 1).

### 3.2. Spatial Distribution of P. falciparum and P. vivax Incidence

Areas with higher incidence of *P. falciparum* among Thai citizens were concentrated along the Thai border with Myanmar and declined over the study period, with the lowest incidence occurring in 2017. The number of sub-districts reporting no *P. falciparum* malaria increased from 77 (47.83%) in 2012 to 145 (90.06%) in 2017 (Appendix A). SEB smoothed incidence rates revealed spatial heterogeneity of annual incidence rates within the study areas. The number of sub-districts with an incidence of <1 per 1000 population per year increased over the study period. For the last two years (2016 and 2017), the number of sub-districts with SEB smoothed incidence rates of *P. falciparum* <1 per 1000 population per year were 58 (36.03%) and 56 (34.78%), respectively. The numbers of sub-districts reporting no *P. falciparum* malaria cases among Thai citizens were 103 (63.98%) and 105 (65.22%) in these years (Figure 2a and Table 2). 

The areas of high incidence of *P. vivax* among Thai citizens were also concentrated along the Thai border with Myanmar and declined 2012–2015, although not as quickly as *P. falciparum*. The number of sub-districts reporting no *P. vivax* malaria cases increased from 66 (40.99%) in 2012 to 111 (68.94%) in 2017 (Appendix A). For the last two years (2016 and 2017), the numbers of sub-districts which showed SEB smoothed incidence rates for *P. vivax* of between 1 and 10 cases per 1000 population per year were 19 (11.80%) and 18 (11.18%). The numbers of sub-districts which had <1 case per 1000 population per year were 71 (44.10%) and 77 (47.83%), and the numbers of sub-districts which had no *P. vivax* cases were 70 (43.48%) and 66 (40.99%) (Figure 2b and Table 2).

Due to the limited data of total number of migrants, the distribution of actual cases of *P. falciparum* and *P. vivax* among migrants (both M1 and M2) were presented spatially in Appendix A. Although direct comparison with the spatial distribution of incidence among the Thai population was not possible, the similar trend could be observed i.e., high numbers of *P. falciparum* cases among migrants occurred between 2012 and 2015 in those sub-districts closest to Myanmar, and still persisted at low numbers from 2016 to 2017. Similar to the trend among the Thais, the reduction of *P. vivax* cases was less prominent and the infections were more widely spread across the region than *P. falciparum*.

### 3.3. Clustered Areas

For both *P. falciparum* and *P. vivax*, the clustered sub-districts of high–high (red) were confined to the north and central parts of the study area, in Tha Song Yang and Um Phang Districts of Tak Province, and Sangkhlaburi District in Kanchanaburi Province. All high–high clustered sub-districts were near to, or shared a border with, the neighboring country (Myanmar), while clustered sub-districts of low–low (blue) were geographically further from the border (Figure 2). 

For clustered areas for *P. falciparum*, the spatial autocorrelation (Global Moran’s *I*) ranked between 0.19 and 0.64 (*P* < 0.006). Eleven sub-districts were defined as local clusters for ≥3 of the 6 years and classified into two groups according to their location (Cluster I and Cluster II). There were four sub-districts in Cluster I; namely, Tha Song Yang, Mea Wa Luang, Mae Song, and Mae Usu and seven sub-districts in Cluster II; namely, Mae Klong, Nong Luang, Um Phang, Mae La Mung, Mae Chan, Lai Wo, and Prang Phle (Figure 3a). 

For clustered areas of *P. vivax*, the spatial autocorrelation ranked between 0.22 and 0.61 (*P* < 0.004). Twelve sub-districts were defined as local clusters for ≥3 out of 6 years and classified into Cluster I and Cluster II. There were six sub-districts in Cluster I; namely, Tha Song Yang, Mea Wa Luang, Mae Song, Mae Usu, Mae Tan, and Mae La and six sub-districts in Cluster II; namely, Mae Klong, Nong Luang, Um Phang, Mae La Mung, Mae Chan, and Nong Lu (Figure 3b). Note that although Nong Lu is not adjacent to the main patch of Cluster II, it was justified to include this sub-district as part of the cluster because of its geographical characteristics. Cluster II covered Umphang District, Tak and Sangkhlaburi District, Kanchanaburi, both of which belong to the same wildlife sanctuaries. The areas in this cluster share similar environmental ecology.

### 3.4. Temporal Distribution and Seasonality

In Cluster I, the number of cases of each of *P. falciparum* and *P. vivax* among Thai citizens showed large peaks in years 2012–2013, followed by significant reductions in 2014 (Figure 4a1 and Figure 5a1). There was a characteristic seasonal pattern in incidence. There were two seasonal peaks: A higher peak, usually in June (one month after the beginning of the rainy season), and a smaller peak, in December to January (Figure 4b1 and Figure 5b1). In contrast, numbers of cases in Cluster II for both *P. falciparum* and *P. vivax* showed the greatest peak in 2013, slowly declining by 2015, with lower incidence in subsequent years (Figure 4a2 and Figure 5a2). The seasonal pattern showed one seasonal peak in Cluster II, with the highest peak usually occurring in June (Figure 4b2 and Figure 5b2). This decrease in cases was also seen in both M1 and M2 migrants (Appendix A).

### 3.5. Changes in Trends

After adjustment for seasonal variation of the time series data and removal of noise, the highest *P. falciparum* incidences were observed in Tha Song Yang (Cluster I) and Mae Chan (Cluster II) sub-districts (Figure 4c1,c2). The regression coefficients indicated that there was a 5% decrease in incidence for Cluster I (IRR 0.95, 95% CI 0.95–0.95) and a 4% decrease in incidence for Cluster II (IRR 0.96, 95% CI 0.96–0.97), for every one additional month of the study period (Table 3). The trend for all individual sub-districts in Cluster I showed a reduction. The most significant decreases were in Tha Song Yang and Mea Usu sub-districts (5% reductions per month), while a slower decrease occurred in Mea Wa Luang and Mae Song sub-districts (3% reductions per month). In Cluster II, the trend for all individual sub-districts was significantly decreased, except for Mae Klong, Lai Wo, and Prang Phle sub-districts. The largest change in trend was a 4% reduction per month in Mae Chan sub-district.

The highest *P. vivax* incidences were observed in Tha Song Yang (Cluster I) and Mae Chan (Cluster II) sub-districts (Figure 5c1,c2). The regression coefficients indicated that there was a 4% decrease in incidence for Cluster I (IRR 0.96, 95% CI 0.96–0.96) and a 3% decrease in incidence for Cluster II (IRR 0.97, 95% CI 0.97–0.97), for every one additional month (Table 3). The most significant decreases in Cluster I were in Mea Usu, Mea Tan, and Mae La sub-districts (5% reductions per month), while a slower decrease was seen in Tha Song Yang sub-district (3% reduction per month). In Cluster II, the trend for all individual sub-districts was a significant decrease, except for Mae Klong sub-district, Umphang district, Tak province.

## 4. Discussion

Our findings showed that 65.22% and 40.99% (for *P. falciparum* and *P. vivax*, respectively) of the sub-districts studied are now already at the incidence level expected for elimination (0 per 1000 people per year), and 34.78% and 47.83% are ready to move to malaria elimination processes (<1 per 1000 people per year) (see Table 2). Malaria has been decreasing over the entire study period in both provinces, however, these decreases were heterogeneous, with small clusters of persistent malaria. 

Seasonal abundance of mosquito vectors [37], and seasonal movement of local people to forests and fields, either to find forest products or to carry out agricultural activities, with overnight stays on either side of the Thai–Myanmar border, can increase the risk of contracting malaria [38]. Seasonal movements among migrants are also common [15]. Previous studies in the region have suggested that *P. falciparum* infections are related to seasonal movement whereas *P. vivax* cases appear to be indigenous (associated with mosquito capture rates and with fewer cases in migrants [14,39]).

Furthermore, from informal interviews and observations conducted in the study areas, the malaria season of Cluster I could be influenced by the seasonal presence of two large cattle markets located in the cluster. Cattle at these markets were imported by Burmese merchants who stayed for several days until all their cattle were sold [40]. These merchants may import malaria to the Thai side of the border. This aggregation of cattle might also attract zoophilic vectors, which subsequently feed opportunistically on humans in the area when their usual food supply is disrupted when the cattle are moved [41]. Cluster II is in a rural, remote area, with many communities far from any public health facilities [14]. Crossing of the international border, through official and unofficial points, is frequent. The relationship between malaria transmission intensity and population movement and migration across the border has been reported previously [20,42,43].

There were a few spatial outliers, sub-districts with low incidence surrounded by sub-districts with high incidence (i.e., “low–high” outliers), identified in this study. One recurrent outlier was a small mountainous sub-district with a few small settlements on the other side of Cluster II. This could be explained by its geographical features, being in a valley area far from the Myanmar border and across mountains from Cluster II, with a small population. Conversely if this outlier had instead been a “high–low” type, it might suggest a role of parasite importation or asymptomatic reservoirs. The large decreases in *P. falciparum* and *P. vivax* incidence rates among Thai citizens began in around 2014, which could be explained partly by an expansion of community access to diagnosis and treatment in Kayin State, Myanmar, on the opposite side of the international border. This area had never previously had adequate malaria services and had long been a reservoir for persistent malaria in the area [20,44]. Now, as this reservoir is being decreased, there might be a corresponding decrease in malaria incidence on the Thai side of the border. Certainly, other external factors such as deforestation and urbanization, which have definitely changed the landscape in this area, and/or meteorological and climatic changes could, as well, influence the distribution of malaria and other diseases like dengue. Further analyses using long-term data would be required to make any causal relationships regarding this phenomenon [14].

Along the Thai–Myanmar border, the availability of Global Fund to fight AIDS, tuberculosis, and malaria support from 2014 to 2017 has boosted interventions to implement community-based services via malaria posts. They have been able to provide free malaria diagnostic testing using RDT, and free treatment, making these services easily accessible to both Thais and migrants. A further Regional Artemisinin Initiative 2 Elimination (RAI2E) grant is supporting Thailand in its efforts to accelerate malaria elimination and target transmission foci between 2018 and 2020. Malaria posts are thus continuing to provide support for hard-to-reach villages along the Thai–Myanmar border [45].

This analysis and findings are useful with regard to: 1) Assessment of both *P. falciparum* and *P. vivax* malaria trends in identified clusters during a period where the malaria interventions were delivered, accounting for seasonality and nationality at a sub-district level and 2) it could further help inform policy decisionmakers with regard to appropriate resource allocation among the cluster and non-cluster areas to achieve malaria elimination, especially given available tools are often limited in poor-resource settings. Regarding the elimination tools, having malaria posts and clinics alone, however, may not be enough to completely eliminate malaria. Vector control practices (including IRS, ITNs, and LLINs) will remain an important strategy for malaria control. Although there are some challenges facing vector control measures due to the biology and behavior of mosquitoes, as well as insecticide resistance [46], such measures remain important. The use of permethrin-impregnated bed nets in this region has previously been associated with a 38% reduction in the number of parasitemia *P. falciparum* and *P. vivax* infections in children, and a 42% decrease in symptomatic episodes [47]. If we are to eliminate malaria quickly, as is probably necessary because of increasing drug resistance, then it will also be necessary to address asymptomatic reservoirs [15].

There were some limitations of this study. Most cases were reported via passive case detection, following the use of RDT in the malaria posts, meaning that asymptomatic cases are likely to be missed. RDTs are also more effective at diagnosing *P. falciparum* than *P. vivax*, meaning that *P. vivax* cases are likely to be missed through RDT diagnosis. Lastly, the issue of data quality is inevitable since the data were derived from the two malaria reporting systems, BOE and BVBD. More detail of data quality assessment has been reported [21].

## 5. Conclusions

This study shows that the incidence of both *P. falciparum* and *P. vivax* malaria cases has been decreasing and has decreased dramatically since 2014. The observed trend highlights the achievement of malaria elimination efforts thus far, including the work of malaria staff and vector control activities. However, the funding and vigilance of the diagnosis and treatment systems must be carefully planned and maintained, even when the malaria burden is heavily reduced. This analysis also showed that there are consistent clusters over time, and given frequent population movement in the area, these clusters could act as reservoirs for the disease. Perhaps one of the main challenges facing elimination programs in this low-transmission setting is maintaining the strong early diagnosis and treatment system, even when malaria cases are very close to zero, and/or preventing re-importation of cases.

## Figures and Tables

**Figure 1 tropicalmed-04-00062-f001:**
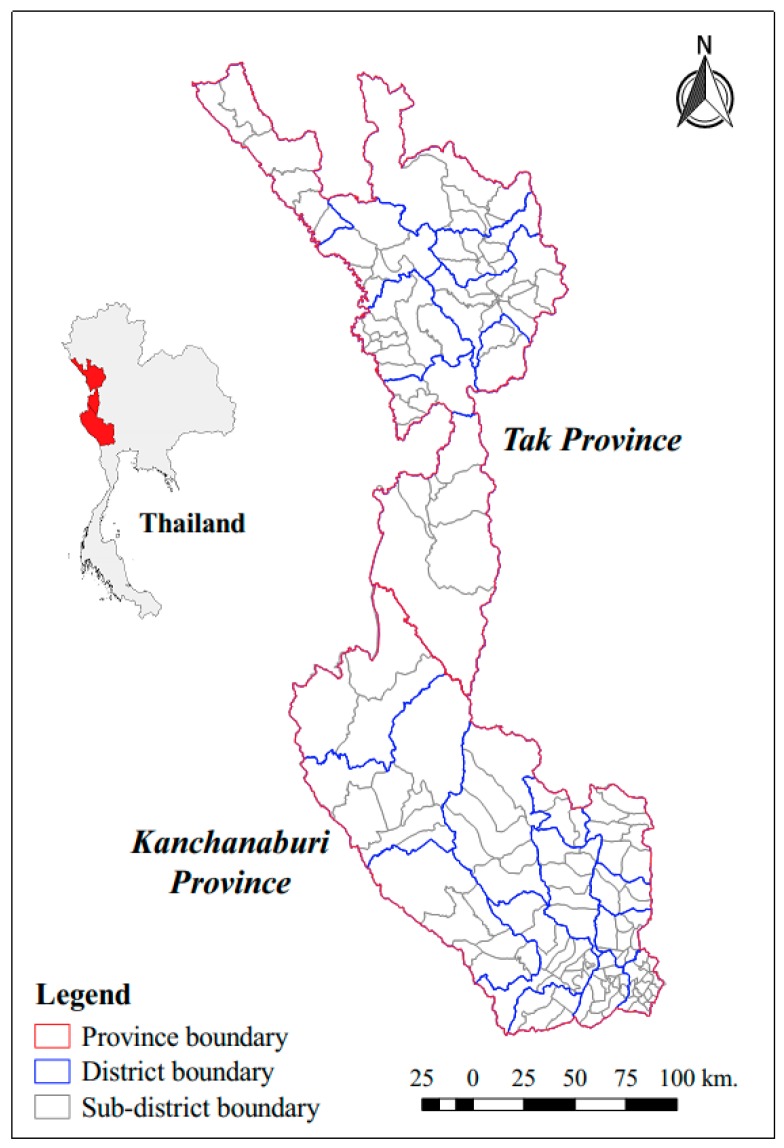
Boundaries of study areas.

**Figure 2 tropicalmed-04-00062-f002:**
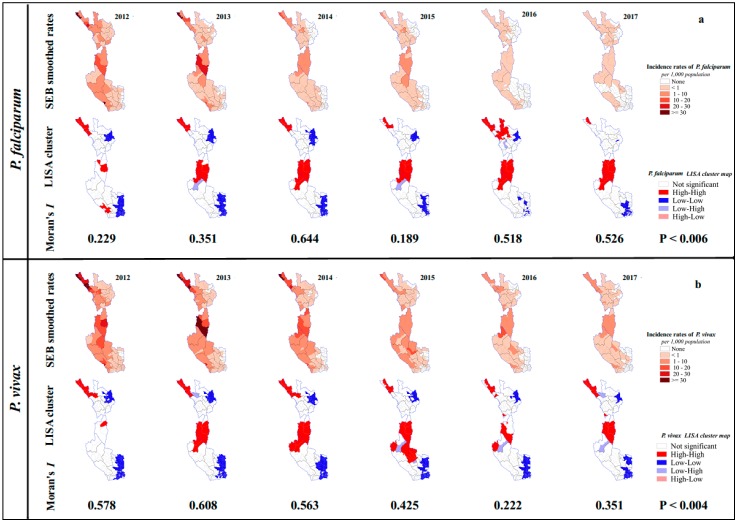
Spatial empirical Bayesian (SEB) smoothed rates, statistically significant local indicators of spatial associations (LISA) cluster map, and Moran’s *I* of *P. falciparum* (**a**) and *P. vivax* (**b**) among Thai citizens.

**Figure 3 tropicalmed-04-00062-f003:**
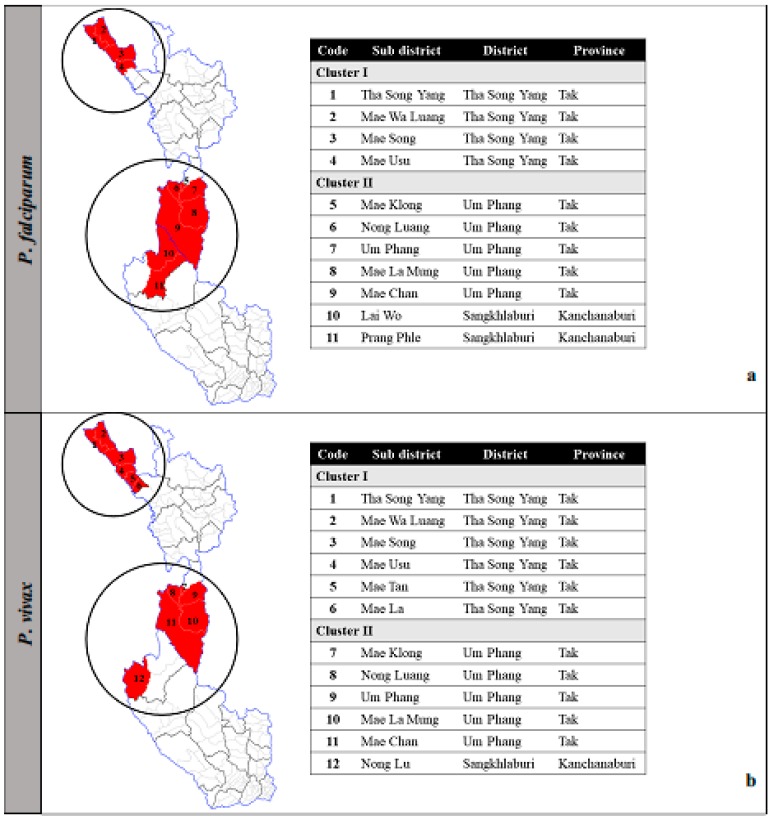
Clustered areas of *P. falciparum* (**a**) and *P. vivax* (**b**) among Thai citizens.

**Figure 4 tropicalmed-04-00062-f004:**
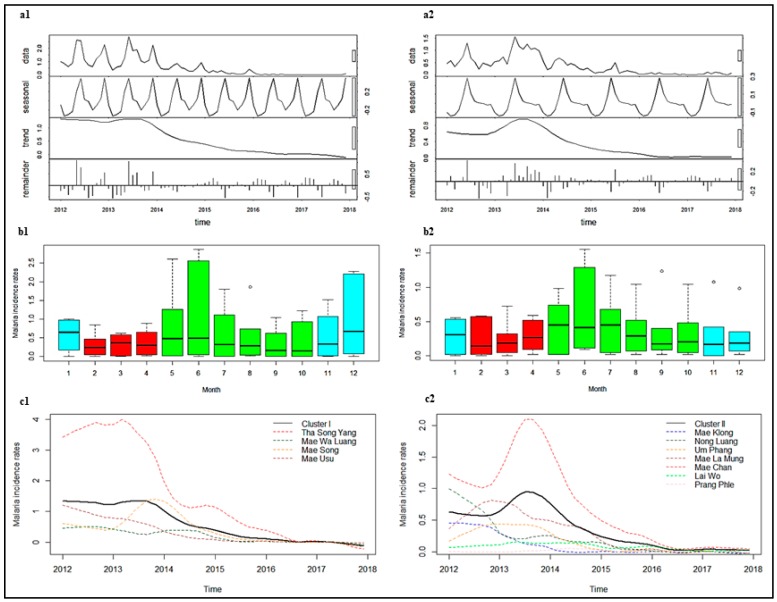
Seasonal Trend decomposition procedure based on Loess or STL (**a**), seasonality (**b**), and trends (**c**) of *P. falciparum* among Thai citizens between 2012 and 2017 in Cluster I (**1**) and Cluster II (**2**). Color of box plot in b1 and b2: Blue represents the cold season, red represents the hot season, and green represents the rainy season. Note: Y-axes do not show equal values, which should be taken into consideration when performing comparisons.

**Figure 5 tropicalmed-04-00062-f005:**
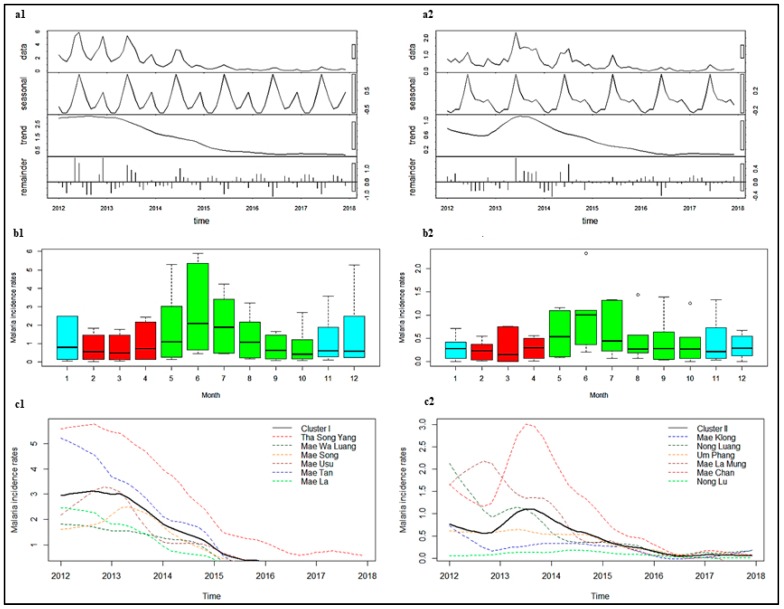
STL (**a**), seasonality (**b**), and trends (**c**) of *P. vivax* among Thai citizens between 2012 and 2017 in Cluster I (**1**) and Cluster II (**2**). Color of box plot in b1 and b2: Blue represents the cold season, red represents the hot season, and green represents the rainy season. Note: Y-axes do not show equal values, which should be taken into consideration when performing comparisons. Incidence of trend in c1 is not exactly equal to zero, see Appendix A.

**Table 1 tropicalmed-04-00062-t001:** Health facilities, *P. falciparum* cases, and *P. vivax* cases reported through Thai health facilities in the study areas.

Characteristics		Tak Province		Kanchanaburi Province	
No. of public hospitals		9		15	
No. of malaria clinics		26		15	
No. of malaria posts		76		41	
		**2012**	**2017**	**2012**	**2017**
*P. vivax* : *P. falciparum* ^†^		1.62	11.45	1.06	13.06
*P. falciparum* cases		4735	90	1719	16
Percent by age	<5 years	9.27	10.00	4.42	6.23
	5 to 14 years	31.83	27.78	18.62	0.00
	15 to 24 years	22.52	18.89	23.79	25.00
	25 to 44 years	24.19	26.67	35.78	37.50
	≥45 years	12.20	16.67	17.39	31.25
By occupation (%)	Agriculture	13.72	14.44	44.56	62.50
	Student	39.78	30.00	21.29	12.50
	Laborer	45.23	54.44	31.41	18.75
	Others *	1.27	1.11	2.73	6.25
By type (%)	Thais	33.04	42.22	30.31	37.50
	M1	33.69	23.33	50.79	56.25
	M2	33.27	34.44	18.90	6.25
Male : Female		1.95	1.37	2.39	1.29
*P. vivax* cases		7660	1041	1820	209
Percent by age	< 5 years	11.87	7.77	7.15	6.22
	5 to 14 years	33.79	31.77	23.53	22.97
	15 to 24 years	22.13	23.03	22.32	23.92
	25 to 44 years	22.00	23.99	28.48	26.79
	≥45 years	10.21	13.44	18.53	20.10
By occupation (%)	Agriculture	11.52	16.31	31.45	42.58
	Student	44.41	36.66	30.07	28.71
	Laborer	42.88	45.20	35.02	24.40
	Others *	1.19	1.82	3.46	4.31
By type (%)	Thais	43.49	43.76	43.32	44.50
	M1	27.85	32.73	41.95	43.54
	M2	28.66	23.51	14.73	11.96
Male : Female		1.55	1.82	2.06	1.90

^†^ The ratios presented a statistically significant change in trend over time for both Tak and Kanchanaburi (*P* = 0.048 and 0.025, respectively), * Other occupations refer to governmental officer, merchant, monk, and nun.

**Table 2 tropicalmed-04-00062-t002:** Number of sub-districts by *P. falciparum* or *P. vivax* incidence rate groups (row) for each year, among Thais. Incidence is per 1000 population per year using SEB smoothed rates.

Incidence Rates	2012	2013	2014	2015	2016	2017
*P. falciparum*						
0	25	45	74	80	103	105
<1	93	93	72	74	58	56
1–10	38	19	14	7	0	0
10–20	2	2	1	0	0	0
20–30	1	1	0	0	0	0
≥30	2	1	0	0	0	0
*P. vivax*						
0	7	24	45	47	70	66
<1	101	88	74	74	71	77
1–10	39	38	33	33	19	18
10–20	6	5	7	7	1	0
20–30	5	3	1	1	0	0
≥30	3	3	1	1	0	0

**Table 3 tropicalmed-04-00062-t003:** Incidence rate ratios (IRR) and 95% confidence interval (CI) of individual sub-districts in Clusters I and II.

Clustered Areas and Individual Sub-Districts	*P. falciparum*		*P. vivax*	
	IRR	(95% CI)	IRR	(95% CI)
Cluster I	0.949	(0.947–0.952)	0.955	(0.953–0.956)
Tha Song Yang	0.950	(0.945–0.953)	0.965	(0.963–0.967)
Mae Wa Luang	0.974	(0.966–0.981)	0.962	(0.957–0.968)
Mae Song	0.965	(0.960–0.969)	0.960	(0.956–0.963)
Mae Usu	0.945	(0.939–0.951)	0.951	(0.947–0.954)
Mae Tan	-	-	0.948	(0.945–0.952)
Mae La	-	-	0.952	(0.947–0.956)
Cluster II	0.962	(0.958–0.965)	0.968	(0.965–0.970)
Mae Klong	0.985	(0.975–0.995)	0.988	(0.979–0.997)
Nong Luang	0.977	(0.968–0.987)	0.973	(0.965–0.981)
Um Phang	0.980	(0.971–0.989)	0.981	(0.974–0.989)
Mae La Mung	0.984	(0.975–0.992)	0.971	(0.963–0.978)
Mae Chan	0.961	(0.957–0.965)	0.965	(0.962–0.968)
Lai Wo	0.991	(0.982–1.000)	-	-
Prang Phle	0.996	(0.986–1.007)	-	-
Nong Lu	-	-	0.986	(0.980–0.992)

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
