# Peer review of "Spatial Heterogeneity and Temporal Trends in Malaria on the Thai–Myanmar Border (2012–2017): A Retrospective Observational Study"

_tropicalmed, 2019, doi:10.3390/tropicalmed4020062_

Round 1

Reviewer 1 Report

This is a really interesting paper from the point of view of the elimination goals of the Ministry of Health and BVBD in Thailand. Interesting analysis using routine data. Overall, it would be good to see more in the discussion about how the MoH/BVBD could use results to further target or intensify strategies in Thailand. More detailed comments below: 

Abstract

Does the elimination date 20206 apply to both Thailand and Myanmar? If not, better to specify which country is aiming for this date.

There are some typos/grammar errors in the abstract that could do with a review e.g.

-          line 29 ‘Thai citizen’ should read ‘Thai citizens’

-          line 30 starting ’65.22% and 40.99% for pf and pv respectively seems like it is beginning the start of the sentence, also the rest of the sentence does not make sense

Results

Table 1: Not clear what the numbers are in the table, might just be a formatting issue e.g. pf cases==4735 – 90? Does that mean it was 4735 in 2012 and reduced to 90 in 2017? It might be worth inserting a separate column for the two years

Table 1 results – It would be good to clarify that results for pf or pv cases are for all classifications of malaria cases (i.e Thai, M1 & M2)

Is there anything to comment re- spatial outliers? Do they contribute any information to understanding underlying epidemiology in areas where they were detected?

Suggest changing the wording in line 327 … ‘become anthropophilic’ to ‘feed opportunistically on humans in the area’ or ‘catholic feeders that are attracted by cattle, but feed opportunistically on humans.’

Minor: Standardise italicising vivax and falc e.g. line 182 italics but line 183 not italicised

Minor: Table 1 – use ‘ : ‘ for the ratio results presented in the table rather than ‘-‘ e.g. P v: pf should be presented as 1.62 : 18.44

Minor: Line 289 – suggest ‘warning’ is changed to ‘note’

Discussion

It would be good to see more discussion around the implication of results for malaria control targeting/strategies for the MoH. For example, could your results be used to target or intensify strategies in certain areas to achieve goals set out in the National strategic documents?

Other than community level control of malaria in Myanmar, are there any other factors that could explain the dramatic decrease in cases seen in Thailand? Are there factors that are external to control e.g. climate that should be considered?

Could authors comment on whether there is anything to be learned from the spatial outliers detected in the analysis? 

Delete lines: 358 – 360

Limitations: It would be good to talk about the quality of data if authors feel that might affect results.

Author Response

Comments and Suggestions for Authors: REVIEWER 1

This is a really interesting paper from the point of view of the elimination goals of the Ministry of Health and BVBD in Thailand. Interesting analysis using routine data. Overall, it would be good to see more in the discussion about how the MoH/BVBD could use results to further target or intensify strategies in Thailand. More detailed comments below:

Abstract

1)      Does the elimination date 2026 apply to both Thailand and Myanmar? If not, better to specify which country is aiming for this date.

No, it is not. It should be 2026 for Thailand and 2030 for Myanmar. We have added this information in the abstract.

“despite a plan for the elimination by the end of 2026 (Thailand) and 2030 (Myanmar)”

2)      There are some typos/grammar errors in the abstract that could do with a review e.g.

·         line 29 ‘Thai citizen’ should read ‘Thai citizens’

We have corrected the text as suggested.

·         line 30 starting ’65.22% and 40.99% for pf and pv respectively seems like it is beginning the start of the sentence, also the rest of the sentence does not make sense

We have adjusted the text to make the sentence more understandable as follow.

“Based on reported incidence, 65.22% and 40.99% of the areas studied were seen to already be at elimination levels for P. falciparum and P. vivax, respectively.”

Results

3)      Table 1: Not clear what the numbers are in the table, might just be a formatting issue e.g. pf cases==4735 – 90? Does that mean it was 4735 in 2012 and reduced to 90 in 2017? It might be worth inserting a separate column for the two years

We have followed the suggestion of the reviewer and inserted column for years 2012 and 2017 as showed in Table 1.

Table 1. Health facilities in the study area, and P. falciparum and P. vivax case information for all malaria cases reported through Thai health facilities.

Characteristics  

Tak   Province

Kanchanaburi   Province

No.   of public hospitals

9

15

No.   of malaria clinics

26

15

No.   of malaria posts

76

41

2012

2017

2012

2017

P.   vivax : P. falciparum

1.62

11.45

1.06

13.06

P.   falciparum cases

4,735

90

1,719

16

Majority   by age (%)

5   to 14

31.79

27.78

25   to 44

34.34

37.50

Majority   by occupation (%)

Student   & Labourer

41.63

48.89

Agriculture

26.35

62.50

Majority   by nationality (%)

Myanmar *

63.76

77.40

Myanmar*

58.58

50.00

Majority   by type (%)

M2

33.69

42.22

M1**

48.74

56.25

Male   : Female

1.95

1.37

2.39

1.29

P.   vivax cases

7,660

1,041

1,820

209

Majority   by age (%)

5   to 14

33.83

31.80

5   to14 & 25 to 44 **

28.52

26.79

Majority   by occupation (%)

Student

44.37

36.50

Student  

32.97

39.24

Majority   by nationality (%)

Myanmar   *

55.07

56.29

Thai   & Myanmar*, **

43.30

52.63

Majority   by type (%)

Thai

43.56

43.80

Thai   & M1**

43.30

44.50

Male   : Female

1.55

1.82

2.06

1.90

The ratios presented a statistical significant change in trend over time for both Tak and Kanchanaburi (P = 0.048 and 0.025, respectively), * P. falciparum and P. vivax cases by nationality of Myanmar were combined from Karen, Mon, and Myanmar cases, ** Majority in 3 out of the 6 years

4)      Table 1 results – It would be good to clarify that results for pf or pv cases are for all classifications of malaria cases (i.e Thai, M1 & M2)

We have followed the suggestion of reviewer and adjusted the text as follow.

“The health facilities in the study area that provided diagnoses (either by microscopy or RDT) and treatment included public hospitals, malaria clinics, and malaria posts [21] for P. faclciparum and P. vivax cases of all classifications (i.e. Thai, M1, and M2) (Table 1).”

5)      Is there anything to comment re- spatial outliers? Do they contribute any information to understanding underlying epidemiology in areas where they were detected?

The consistent spatial outlier in Figure 2 (“Low-Highs” cluster) is Prang Phe sub-district. This sub-district had low incidence while their neighbors had high incidence. Prang Phe area is mostly mountains with a few small settlements on the other side of the mountains from Umphang (which had high incidence). This may just be an effect of the shape of the district. It is bordering places with high incidence, but really it is isolated from those places too - it is in a valley area a little far from the Myanmar border and across mountains from Umphang. We have no information of health services of this area, but it almost looks like malaria (both Pf and Pv) decreased there faster than in its neighbors - which could be related to health services, or small population that is geographically sheltered from its neighbors with higher incidence. If these outliers were instead "High-Lows" then it might suggest a role of parasite importation or asymptomatic reservoirs. We have added a statement in the discussion section to address this.

6)      Suggest changing the wording in line 327 … ‘become anthropophilic’ to ‘feed opportunistically on humans in the area’ or ‘catholic feeders that are attracted by cattle, but feed opportunistically on humans.’

We have adjusted the text follow the suggestion of reviewer from “become anthropophilic” to “feed opportunistically on humans in the area”

7)      Minor: Standardise italicising vivax and falc e.g. line 182 italics but line 183 not italicized

We have adjusted the text follow the suggestion of reviewer.

8)      Minor: Table 1 – use ‘ : ‘ for the ratio results presented in the table rather than ‘-‘ e.g. P v: pf should be presented as 1.62 : 18.44

We calculated ratio PV: PF year by year and have separated for two columns to show them in year 2012 and 2017 in Table 1.

9)      Minor: Line 289 – suggest ‘warning’ is changed to ‘note’

We have adjusted the text follow the suggestion of reviewer.

Discussion

10)  It would be good to see more discussion around the implication of results for malaria control targeting/strategies for the MoH. For example, could your results be used to target or intensify strategies in certain areas to achieve goals set out in the National strategic documents?

We have followed the reviewer’s suggestion as follow:

“This analysis and findings are useful with regard to: 1) assessment of both P. falciparum and P. vivax malaria trends in identified clusters during a period where the malaria interventions were delivered, accounting for seasonality and nationality at a sub-district level and 2) it could further help inform policy decision makers with regard to appropriate resource allocation among the cluster and non-cluster areas to achieve malaria elimination, especially given available tools are often limited in poor-resource settings.”

11)  Other than community level control of malaria in Myanmar, are there any other factors that could explain the dramatic decrease in cases seen in Thailand? Are there factors that are external to control e.g. climate that should be considered?

We think deforestation and urbanization have definitely changed the landscape in this area, and also the distribution of malaria (and other diseases like dengue). It is really difficult to make causal relationships with this though because you need good long-term data [1]. This wouldn't be enough to explain the decrease around 2014 but could explain general spatial patterns. It is possible that there are meteorological and climatic changes too - but there really isn't a long-enough time series to say whether or not these are changing drastically over time. El Nino cycles might be something to look at but it isn't clear that they necessarily lead to less or more malaria, especially in places that have both a wet season and a cold season peak in malaria.

Some sentences have been added in the discussion:

“Certainly other external factors such as deforestation and urbanization, which have definitely changed the landscape in this area, and/or meteorological and climatic changes could as well influence the distribution of malaria and other diseases like dengue. Further analyses using long-term data would be required to make any causal relationships regarding this phenomenon [14].”

Reference

1. Parker, D.M.; Carrara, V.I.; Pukrittayakamee, S.; McGready, R.; Nosten, F.H. Malaria ecology along the Thailand–Myanmar border. Malar. J. 2015, 14, 388, doi:10.1186/s12936-015-0921-y.

12)  Could authors comment on whether there is anything to be learned from the spatial outliers detected in the analysis?

We have added the text to the discussion as follow:

“There were a few spatial outliers, sub-districts with low incidence surrounded by sub-districts with high incidence (i.e. “Low-High” outliers), identified in this study. One recurrent outlier was a small mountainous sub-district with a few small settlements on the other side of Cluster II. This could be explained by its geographical features, being in a valley area far from the Myanmar border and across mountains from Cluster II, with a small population. Conversely if this outlier had instead been a “High-Low” type, it might suggest a role of parasite importation or asymptomatic reservoirs.”

13)  Delete lines: 358 – 360

We have removed the last paragraph in the discussion section.

14)  Limitations: It would be good to talk about the quality of data if authors feel that might affect results.

We agreed with the reviewer on this point and therefore mentioned this in the limitation.

“Lastly, the issue of data quality is inevitable since the data were derived from the two malaria reporting systems, BOE and BVBD. More detail of data quality assessment has been reported [21].

Reviewer 2 Report

Overall Comments:

This work conducts a spatio-temporal analysis of routinely collected malaria data from the Thai-Myanmar border area. In total 161 sub-districts spanning 5 years were analyzed. The authors identify 2 main clusters of malaria which exhibited an overall decreasing incidence over time. Overall, this manuscript is well written, applies a robust analysis and is a good use of existing surveillance data to get a better understanding of the malaria epidemiology in this setting . However, I have the following specific comments that should be addressed prior to recommending for publication.

Specific Comments:

-      The authors justify their focus on Thai citizens, but I’m curious to why the authors excluded the M1 cases, or individuals who had been in country for >6 months. Presumably, these infections would also likely be contracted locally and be reflective on indigenous transmission. Given that most cases were observed in this population it would be good so see if this impacts the spatial-temporal patterns observed. The results in figure S2 confirm this  as transmission patterns mirror those observed in Thai nationals

-      Table 1: I find the information confusion/misleading as it is currently presented. What is the Pv:Pf ratio? Is this the range of rations between sub-districts? Is the Pf cases line the difference from 2012-2017? Or cluster level difference? Similarly, the range between years in the parentheses seems to be masking a lot of data and the annual trends throughout the time period should be included to better assess the data. This could easily be done without making the table unwieldly and would be a more effective presentation of data.

-      Pv clusters: I see 3 separate clusters in figure 3b. What is the rationale for including the 2 extra sub-districts as part of the larger cluster in cluster II? 

Author Response

Comments and Suggestions for Authors: REVIEWER 2

Overall Comments:

This work conducts a spatio-temporal analysis of routinely collected malaria data from the Thai-Myanmar border area. In total 161 sub-districts spanning 5 years were analyzed. The authors identify 2 main clusters of malaria which exhibited an overall decreasing incidence over time. Overall, this manuscript is well written, applies a robust analysis and is a good use of existing surveillance data to get a better understanding of the malaria epidemiology in this setting. However, I have the following specific comments that should be addressed prior to recommending for publication.

 Specific Comments:

1)      The authors justify their focus on Thai citizens, but I’m curious to why the authors excluded the M1 cases, or individuals who had been in country for >6 months. Presumably, these infections would also likely be contracted locally and be reflective on indigenous transmission. Given that most cases were observed in this population it would be good so see if this impacts the spatial-temporal patterns observed.

We thank the reviewer for bringing this important point up for clarification. Inclusion of migrants M1 and M2 has been discussed quite a lot when we wrote the paper. We excluded the analysis of migrants initially because the estimation of total number was not reliable, making the calculation of incidence inaccurate. By definition, migrants move around more than the others. This makes it hard to really keep track of the total number. Moreover, the fact that they are more likely to move back and forth across the border frequently, we would be less certain about where they may have acquired an infection when compare to Thai population. In addition, migrants do not always cross boundaries legally thus the true number of them is not possible.

However, we would like to propose presenting the actual malaria cases among migrants, instead of the incidence rate shown among Thai citizen. We have added the map of P. falciparum and P. vivax cases among M1 and M2 migrants in Supplementary Figure S2 and mentioned those distributions briefly in the main text.

“Due to the limited data of total number of migrants, the distribution of actual cases of P. falciparum and P. vivax among migrants (both M1 and M2) were presented spatially in Supplementary Figure S2. Although direct comparison with the spatial distribution of incidence among the Thai population was not possible, the similar trend could be observed i.e. high numbers of P. falciparum cases among migrants occurred between 2012 and 2015 in those sub-districts closest to Myanmar, and still persisted at low numbers from 2016 to 2017. Similar to the trend among the Thai, the reduction of P. vivax cases were less prominent and the infections were more widely spread across the region than P. falciparum.”

2)      Table 1: I find the information confusion/misleading as it is currently presented. What is the Pv:Pf ratio? Is this the range of rations between sub-districts? Is the Pf cases line the difference from 2012-2017? Or cluster level difference? Similarly, the range between years in the parentheses seems to be masking a lot of data and the annual trends throughout the time period should be included to better assess the data. This could easily be done without making the table unwieldly and would be a more effective presentation of data.

We have adjusted the Table 1 to include several figures which changed during 2012 and 2017. This includes the PV/PF ratio which changed significantly over time. The analysis of trend has been added to the footnote below the table.

3)       Pv clusters: I see 3 separate clusters in figure 3b. What is the rationale for including the 2 extra sub-districts as part of the larger cluster in cluster II? 

We decided to include the one extra sub-district (Nong Lu) as part of the cluster II because of its geographical characteristics. Cluster II covered Umphang District, Tak and Sangkhlaburi District, Kanchanaburi, both of which belong to the Thungyai-Huai Kha Khaeng Wildlife Sanctuaries. The areas in this cluster share similar environmental ecology. In the same way, the two provinces in the cluster are adjacent to Kayin state of Myanmar.  Having two clusters for the analysis of PV data would simplify the comparison with PF.

Reviewer 3 Report

Spatial heterogeneity and temporal trends in malaria on the Thai-Myanmar border (2012-2017): a retrospective observational study

Thank you allowing me to review this manuscript. It is very interesting study and is timely in describing the spatial and temporal trends in malaria incidence on the Thai-Myanmar border. In areas approaching elimination targeting interventions at the highest risk areas can become difficult as cases become fewer and more difficult to identify and describe. The methods that the authors use are appropriate and account for these low case counts. I have no major comments that need addressed in this manuscript. It is well written, used appropriate statistical and spatial methods in their analyses, and the results support their conclusions. It is also timely and displays a public health importance. I have a few minor comments listed below.

Introduction

At the end of the third paragraph the authors list the interventions that are being implemented in the area. It would be helpful to have the timing (year) of each of the interventions to be able to later link them to the temporal trends in incidence.

Materials and Methods

In the ‘Malaria data’ section where the diagnostic methods are described, the specifics of the brand of the RDT that was used, the antigens they detect, and their sensitivity and specificity could be added to better understand the potential for cases being missed at this level.

Results

The tables and figures are really well laid out to present the results.

In the ‘Changes in trends’ section the confidence intervals are really tight (e.g. .95-.95). This may be something to look into to confirm.

Discussion

The last paragraph is placeholder/instructional text and should be removed.

The authors link the decreases in incidence to the implementation of interventions. A causal model to investigate this wasn’t presented, so this shouldn’t be made directly. The exact timing of the interventions would help make a hypothesis here, since declines were pretty early in the study period.

The authors should consider including some mention about how their findings could be used to inform delivery of the interventions or inform policy as these areas approach elimination.

Author Response

Comments and Suggestions for Authors: REVIEWER 3

Spatial heterogeneity and temporal trends in malaria on the Thai-Myanmar border (2012-2017): a retrospective observational study

Thank you allowing me to review this manuscript. It is very interesting study and is timely in describing the spatial and temporal trends in malaria incidence on the Thai-Myanmar border. In areas approaching elimination targeting interventions at the highest risk areas can become difficult as cases become fewer and more difficult to identify and describe. The methods that the authors use are appropriate and account for these low case counts. I have no major comments that need addressed in this manuscript. It is well written, used appropriate statistical and spatial methods in their analyses, and the results support their conclusions. It is also timely and displays a public health importance. I have a few minor comments listed below.

Introduction

1)      At the end of the third paragraph the authors list the interventions that are being implemented in the area. It would be helpful to have the timing (year) of each of the interventions to be able to later link them to the temporal trends in incidence.

We have followed the reviewer’s suggestion as follow:

“In order to achieve the goal, several malaria control and elimination strategies have been implemented in endemic areas along the Thai-Myanmar border i.e. insecticide‐treated nets (ITNs), long-lasting insecticide-treated bed nets (LLINs) (adopted since 2008 [12]), indoor residual insecticide spraying (IRS) (adopted since 1953 [12]), rapid diagnostic tests (RDTs) (adopted since 2001-2002 [13] and at the same time as malaria post in 2014 [11]) and ACTs (adopted since 1995 [12] and intensively implemented in both Thailand and Myanmar border from 2014 [11, 13-15])..

Materials and Methods

2)      In the ‘Malaria data’ section where the diagnostic methods are described, the specifics of the brand of the RDT that was used, the antigens they detect, and their sensitivity and specificity could be added to better understand the potential for cases being missed at this level.

 We have followed the advice of the reviewer and adjusted the text as follow:

“Recently, RDT used in Thai-Myanmar border is CareStart™ Malaria HRP2/pLDH (Pf/pan) Combo which is based on detection of malaria antigens such as histidine-rich protein 2 (HRP2) for P. falciparum and pan malaria-specific antigen (pLDH) for all other malarial species. The sensitivity and specificity of the test reported were 97.3% and 94.1%, respectively [22].”

Results

The tables and figures are really well laid out to present the results.

3)      In the ‘Changes in trends’ section the confidence intervals are really tight (e.g. .95-.95). This may be something to look into to confirm.

We added into three decimal point in both IRR and 95% CI in Table 3 in order to show clearly range of values.

Discussion

4)      The last paragraph is placeholder/instructional text and should be removed.

We have removed the last paragraph in the discussion section.

5)      The authors link the decreases in incidence to the implementation of interventions. A causal model to investigate this wasn’t presented, so this shouldn’t be made directly. The exact timing of the interventions would help make a hypothesis here, since declines were pretty early in the study period.

We have adjusted the text such that the causal effect cannot be assumed. Also some other external factors were mentioned in the discussion now.

6)      The authors should consider including some mention about how their findings could be used to inform delivery of the interventions or inform policy as these areas approach elimination.

We have added the implication of findings in policy decision strategic planning in the discussion section.

“This analysis and findings are useful with regard to: 1) assessment of both P. falciparum and P. vivax malaria trends in identified clusters during a period where the malaria interventions were delivered, accounting for seasonality and nationality at a sub-district level and 2) it could further help inform policy decision makers with regard to appropriate resource allocation among the cluster and non-cluster areas to achieve malaria elimination, especially given available tools are often limited in poor-resource settings.”

Round 2

Reviewer 2 Report

This version of the manuscript is much improved. I however still have a couple of comments that would strengthen this work:

1) LL123/124: The authors should clarify that the sensitivity/specificity statistics presented are for Pf (as the authors note in the discussion, the performance for Pv are worse but is best available). The authors should also include the limit of detection for which their statistics are relevant, and potentially noting that in symptomatic malaria, the focus of this work, this parasite density is usually exceeded in clinically detectable cases. The same research group has shown that the performance of these diagnostics in field settings to identify asymptomatic infections are much lower than what is reported.

2) The revised table 1 is much improved. However, I'm left wondering what the prevalence is of the other groupings. The most frequent occupation group etc. is helpful, but without reporting the other categories, the story is incomplete. For example, if the 'second place' group is only a few percentage points difference, then this is also an important risk group. Similarly, knowing which groups are very low risk is also helpful for planning any targeted intervention (so efforts aren't wasted targeted these populations unnecessarily). 

3) The authors have argued the grouping of cluster 2 on the basis of the 2 discrete units having similar ecology. This is not presented in the paper nor is there evidence provided for this decision. Similarly, what is the impact on the resulting trends of including/excluding this 'non-hot' area is their subsequent analysis? If no impact, then the grouping is justified, but this evidence should be available to readers.

Author Response

Comments and Suggestions for Authors

This version of the manuscript is much improved. I however still have a couple of comments that would strengthen this work:

Thank you very much. We are grateful to all useful comments in the first round.

1) LL123/124: The authors should clarify that the sensitivity/specificity statistics presented are for Pf (as the authors note in the discussion, the performance for Pv are worse but is best available).

The sentence was updated.

“The sensitivity and specificity of the test for P. falciparum reported were 97.3% and 94.1%, respectively [25]

The authors should also include the limit of detection for which their statistics are relevant, and potentially noting that in symptomatic malaria, the focus of this work, this parasite density is usually exceeded in clinically detectable cases. The same research group has shown that the performance of these diagnostics in field settings to identify asymptomatic infections are much lower than what is reported.

A sentence was added to the RDT information.

“The limit of detection of RDTs was estimated to be around 100 parasites/μL, the densities at which, in low transmission settings, malaria infections are commonly presented with the signs and symptoms.”

2) The revised table 1 is much improved. However, I'm left wondering what the prevalence is of the other groupings. The most frequent occupation group etc. is helpful, but without reporting the other categories, the story is incomplete. For example, if the 'second place' group is only a few percentage points difference, then this is also an important risk group. Similarly, knowing which groups are very low risk is also helpful for planning any targeted intervention (so efforts aren't wasted targeted these populations unnecessarily).

The table was updated accordingly.

3) The authors have argued the grouping of cluster 2 on the basis of the 2 discrete units having similar ecology. This is not presented in the paper nor is there evidence provided for this decision.

The sentence was added.

“Note that although Nong Lu is not adjacent to the main patch of Cluster II, it was justified to include this sub-district as part of the cluster because of its geographical characteristics. Cluster II covered Umphang District, Tak and Sangkhlaburi District, Kanchanaburi, both of which belong to the same wildlife sanctuaries. The areas in this cluster share similar environmental ecology.”

Similarly, what is the impact on the resulting trends of including/excluding this 'non-hot' area is their subsequent analysis? If no impact, then the grouping is justified, but this evidence should be available to readers.

We would like to clarify that not ALL subdistricts in that circle were included for further analysis, only the hotspot sub-districts were. That is if they were statistically significant hotspots for 3 or more years. Thus there were six sub-districts in Cluster II; namely Mae Klong, Nong Luang, Um Phang, Mae La Mung, Mae Chan, and Nong Lu. Note that we have updated Figure 3b to make the numbers seen clearer in this round.

Round 3

Reviewer 2 Report

I have no additional comments and am happy to recommend this work for publication.